# Response of Cyanic and Acyanic Lettuce Cultivars to an Increased Proportion of Blue Light

**DOI:** 10.3390/biology11070959

**Published:** 2022-06-24

**Authors:** Laura Cammarisano, Oliver Körner

**Affiliations:** Leibniz-Institute of Vegetable and Ornamental Crops (IGZ), Theodor-Echtermeyer-Weg 1, 14979 Großbeeren, Germany; koerner@igzev.de

**Keywords:** vertical farming, lettuce cultivars, anthocyanin, light spectral quality, stomata

## Abstract

**Simple Summary:**

Indoor crop cultivation systems such as vertical farms or plant factories necessitate artificial lighting. The composition of light quality (i.e., spectral composition) within these systems plays a key role in crop growth and development. Conflicting results on the effects of the light spectrum reported for different plant species and cultivars confirm the specificity of light requirements and the dependency on interacting factors. In this paper, we have therefore investigated how a certain light quality (light with a high share of blue) affects photosynthetic and morphological parameters in two contrasting lettuce cultivars (red and green leaves) with a similar leaf shape and phenotype. The results obtained suggest the occurrence of distinctive morpho-physiological adaptive strategies in green and red pigmented lettuce cultivars to adapt to the higher proportion of blue light environment.

**Abstract:**

Indoor crop cultivation systems such as vertical farms or plant factories necessitate artificial lighting. Light spectral quality can affect plant growth and metabolism and, consequently, the amount of biomass produced and the value of the produce. Conflicting results on the effects of the light spectrum in different plant species and cultivars make it critical to implement a singular lighting solution. In this study we investigated the response of cyanic and acyanic lettuce cultivars to an increased proportion of blue light. For that, we selected a green and a red leaf lettuce cultivar (i.e., ‘Aquino’, CVg, and ‘Barlach’, CVr, respectively). The response of both cultivars to long-term blue-enriched light application compared to a white spectrum was analyzed. Plants were grown for 30 days in a growth chamber with optimal environmental conditions (temperature: 20 °C, relative humidity: 60%, ambient CO_2_, photon flux density (PFD) of 260 µmol m^−2^ s^−1^ over an 18 h photoperiod). At 15 days after sowing (DAS), white spectrum LEDs (WW) were compared to blue-enriched light (WB; λ_Peak_ = 423 nm) maintaining the same PFD of 260 µmol m^−2^ s^−1^. At 30 DAS, both lettuce cultivars adapted to the blue light variant, though the adaptive response was specific to the variety. The rosette weight, light use efficiency, and maximum operating efficiency of PSII photochemistry in the light, F_v_/F_m_’, were comparable between the two light treatments. A significant light quality effect was detected on stomatal density and conductance (20% and 17% increase under WB, respectively, in CVg) and on the modified anthocyanin reflectance index (mARI) (40% increase under WB, in CVr). Net photosynthesis response was generally stronger in CVg compared to CVr; e.g., net photosynthetic rate, P_n_, at 1000 µmol m^−2^ s^−1^ PPFD increased from WW to WB by 23% in CVg, compared to 18% in CVr. The results obtained suggest the occurrence of distinct physiological adaptive strategies in green and red pigmented lettuce cultivars to adapt to the higher proportion of blue light environment.

## 1. Introduction

Indoor vertical farms (IVFs), also called plant factories with artificial lighting (PFALs) as, e.g., described by [1], are completely closed and continuous production systems for crops that utilize vertical space, a controlled environment, and artificial light [2,3,4]. These innovative production systems represent a good solution for producing food locally and without climate impact while concurrently contributing to lessening transportation and food waste and strengthening quality food security [5]. Indeed, crop food production in such systems can be finely tuned to control yield and, peculiarly, morphological and nutritional quality [6], making it possible to increase produce value. For example, applying blue light (B) in an appropriate proportion to other wavelengths, especially to red light (R), enhances crop quality by stimulating the accumulation of secondary compounds [7,8]. In some cases, blue light has also been reported to have positive effects on stomatal conductance, g_s_, and photosynthesis [9,10]. However, conflicting results on the optimal proportion of blue light are reported in the literature, especially between different plant species and cultivars, as in the case of lettuce [11,12,13]. Differing responses actuated by distinct lettuce cultivars may originate from variety-specific characteristics, including morphological and physiological features, such as plant architecture, leaf pigment pool, and stomata traits. These characteristics can account for peculiar light absorption, light use, photosynthetic rate, biomass accumulation, and secondary metabolite content, which can make some cultivars more or less suitable for a certain environment [14,15]. For instance, some cultivars may be more sensitive, or less tolerant, to temperature stress. Cold stress is reported as an effective stimulant for the accumulation of secondary metabolites in a wide range of plants [16], although it could also inhibit growth. In the case of [17], despite cold stress being effective in increasing the accumulation of antioxidants in both the investigated *Capsicum* cultivars, it elicited a contrasting effect on growth. The characterization of cultivar-specific traits to definite conditions can be very relevant for improving both production efficiency and produce quality. Additionally, information on such peculiar properties may be valuable for breeding practices aiming at creating new, more resilient, and nutrient-rich cultivars.

For instance, the secondary metabolite pool, including carotenoids, seems to vary significantly between lettuce cultivars, mainly based on leaf color [18], and this characteristic pool can make cultivars more or less suitable for a certain environment. While red leaf (cyanic) lettuce cultivars are reported to be more plastic to light intensity and spectral composition [19], green (acyanic) cultivars seem to be more sensitive and less capable of adapting to and overcoming the potential light stress [20]. The main reason behind such varied behavior may be the distinct pool of pigments characteristic of red leaves, i.e., an abundance of anthocyanins, lower chlorophyll a:b ratios, and a smaller xanthophyll cycle pool [21]. Thanks to the anthocyanin preventive (through shielding underlying chlorophylls from green, and blue in a minor percentage, photons) and defensive (through antioxidant capacity) functions, red pigmented plants have a higher photoprotective capacity and are considered to cope better with high light [20,22]. Carotenoids, which are generally more abundant in acyanic leaves, have a similar photoprotective role to that of anthocyanins in cyanic leaves, though, due to lacking the shielding function, they are not as effective in reducing the energy load of the photosynthetic apparatus, and thus there is a higher probability of damage [23]. In addition to leaf pigments, various examples of variety-specific responses implemented to adjust to the surrounding environmental conditions have been identified in the literature [24,25]. Distinct responses can also be attributed to cultivar-specific behaviors, such as differences in stomata responses [26]. For instance, cultivars which tend to increase stomata density and, consequently, evapotranspiration, could be more productive in warmer conditions [27].

The great network of adaptive mechanisms that helps the plant adjust to the light environment acts at multiple levels and with different timing [28,29]. Early responses, including adjustments in leaf angle, are beneficial to mitigate the stress effect and prevent the onset of damage. Longer-term adaptation is established when the adverse condition persists, becoming the new standard, and through physiological strategies, e.g., changes to stomata density, allowing plant growth with more or less repercussions [30].

Our aim was to investigate the cultivar-specific adaptive response of differently pigmented lettuce to higher energy light. We hypothesize that (1) alternative and analogous adaptive strategies develop in cyanic and acyanic lettuce cultivars in response to long-term higher energy radiation, applied as blue-enriched white spectrum for 15 days, (2) allowing for regular growth through altered physiology. Therefore, we conducted experiments selecting two lettuce cultivars with a similar architecture and leaf shape, mainly differing in leaf pigmentation, and investigated the cultivar-specific response to light spectral quality (i.e., ‘Aquino’ as acyanic and ‘Barlach’ as cyanic). To assess the impact of light quality on these two contrasting lettuce cultivars, next to destructive observations, non-destructive measurements including light-adapted chlorophyll a fluorescence, stomatal conductance, stomatal traits, photosynthetic rate, and leaf optical properties were taken.

## 2. Materials and Methods

### 2.1. Experimental Design

One experiment with six replications was conducted with two light treatments (white-blue light (WB), and white light control (WW)) and two lettuce cultivars (green leaf lettuce ‘Aquino’ cv. (CVg), red leaf lettuce ‘Barlach’ cv. (CVr), Rijk Zwaan, The Netherlands), resulting in four experimental treatments (WW_CVg; WB_CVg; WW_CVr; WB_CVr).

The two lettuce cultivars were chosen based on their similar plant architecture and leaf shape. The experiment was performed in four separate compartments in a climate-controlled growth chamber (2.40 × 3.85 × 2.20 m; York) at the Leibniz-Institute of Vegetable and Ornamental Crops (Grossbeeren, Germany). The experiment was planned as a split plot block design, with light treatment as the main plot and cultivars as the sub-plot. At three time points (6 October, 10 and 29 November 2021) six young plants (15 days after sowing, DAS) from each of the two cultivars were randomly placed between 10:00 a.m. and 11:00 a.m. in four separated cultivation areas, i.e., 12 plants per shelf, (for technical description see below), each considered as one statistical replication. This resulted in a total of six replications, i.e., three time points with two spatial replications each time.

### 2.2. Plant Cultivation and Light Treatments

Seeds from both lettuce cultivars were germinated in peat plugs (3 cm, Jiffy Growblocks, Jiffy Growing Solutions, Zwijndrecht, The Netherlands) for the first replication and stone-wool cubes (4 cm, Rockwool^®^, Grodan, Roermond, The Netherlands) for the second and third replications. After 24 h in the dark and refrigerated cool conditions (4 °C), seeds were moved (in the morning) to the growth chamber, under white light (260 µmol m^−2^ s^−1^ for an 18 h photoperiod) with controlled temperature (20 °C; day and night) and relative humidity (60%; day and night). After seedling establishment (at 15 DAS, with 5 leaves > 1 cm), the young plants including roots and substrate were inserted into stone-wool cubes (10 cm, Rockwool^®^, Grodan, The Netherlands) and allocated to the different compartments of the growth chamber, where light treatments were applied for the next consecutive 15 days. WW was compared to WB throughout the period (spectral composition, see Table 1). The light intensity of the two treatments was comparable (i.e., similar), in terms of PFD (263.25 ± 6.30 and 259.10 ± 8.46 µmol m^−2^ s^−1^ for WW and WB, respectively) and PPFD (243.03 ± 5.74 and 247.61 ± 8.05 µmol m^−2^ s^−1^ for WW and WB, respectively), between the light treatments.

Each of the two light treatments was replicated in two compartments at the same time, and each was replicated three times (see above). In every compartment, light was applied with two dimmable 8-channel LED lamps (LightDNA8, Valoya, Finland) adjusted to homogenous light distribution at the growth surface. The irradiance and light spectral composition of the treatments were measured using a PAR spectrometer (UPRtek PG200N, 350–800 nm; UPRtek Corp., Taiwan) at the beginning of each trial at each plant canopy level. Figure 1 illustrates the averaged measured light spectra of WW and WB.

Irrigation was provided four times during the light period (evenly distributed over the light period, i.e., 4:00 a.m.–10:00 p.m., with an irrigation event of 1 min) with nutrient solution prepared for lettuce (EC: 1.9 dS m^−1^, pH: 5.5–6) [31]. EC, pH, and water consumption were controlled weekly. Each cultivation area was separately irrigated and its microclimate individually monitored every 15 min (Tinytag Ultra 2, Gemini Data Loggers, Chichester, UK).

The growth chamber was equipped with racking systems, each containing two layers (1.3 × 0.50 m each). Only evenly irradiated areas of the shelves were used for cultivation (0.70 × 0.30 m) of the twelve plants (i.e., 66.67 plants m^−2^). For determination of the transpiration rate, the area contained two empty stone-wool cubes. Each plant was kept in the same position for the whole experimental period, and replicated in two planned distributed blocks of 6 plants each (2/cultivar) to have a more homogeneous representation of the environmental variability within the growth area. The two empty stone-wool cubes were placed in each compartment to account for water evaporation. The growth area, including the stone-wool cubes, was covered with a white plastic sheet to reduce evaporation.

### 2.3. Non-Destructive Measurements

#### 2.3.1. Plant Physiology and Morphology

For analyzing responses to light treatment in the investigated lettuce cultivars, various physiological measurements were performed on different plants at 30 DAS. Samples for the measured physiological and morphological parameters were preselected, based on their position, to gather a population representative of the potential environmental variability, e.g., border effects across the growth area used. For leaf measurements, the same leaf number (counted from the bottom) was employed for different plants and leaf numbers ranging between 11 and 14 were chosen. A greater leaf number was selected for CVg compared to CVr due to distinct plant development. Measurements of different cultivars and light treatments were alternated. Stomatal measurements, e.g., stomatal traits or stomatal conductance, were always measured between 10:30 and 11:30 a.m.

#### 2.3.2. Light-Adapted Imaging Chlorophyll a Fluorescence

Chlorophyll a fluorescence was measured on light-adapted plants using the modulated fluorescence imaging apparatus FluorCam (PSI, Czech Republic). Fluorescence quenching analysis protocol [32] was performed on two plants per replicate (n = 2; N = 48) and manual standard size mask selection was used to define an equal area size to be measured.

#### 2.3.3. Light Response Curve and Leaf Photosynthetic Rate Estimation

At each timely replication, two plants of each replicate and treatment (n = 2, N = 48) were used to measure the photosynthetic light response curve (at PPFD courses of 260, 100, 50, 0, 260, 600, and 1200 µmol m^−2^ s^−1^) (LI-COR 6400XT, Licor Biosciences, Lincoln, NE, USA). To minimize gradients between the growth chamber ambient conditions and inside the cuvette of the gas exchange system, the sample CO_2_ concentration, relative humidity, and leaf temperature inside the cuvette were set to 400 µmol mol^−1^, 60%, and 22 °C, respectively. For these measurements, leaf number 14 and leaf number 12 (counted from the first unfolded leaf) were used for CVg and CVr, respectively. Leaf net photosynthesis (P_n_, µmol [CO_2_] m^−2^ s^−1^) measurements were fitted to the non-rectangular hyperbolic function [33] and the exponential light response curve (P_n_ = P_g,max_ (1 − exp [(−ε PPFD)/P_g,max_] − R_d_) to estimate chemical light use efficiency (ε, mol CO_2_ mol^−1^ photons), the theoretical maximum leaf net and gross photosynthesis values (P_n,max_ or P_g,max_, µmol [CO_2_] m^−2^ s^−1^), and leaf dark respiration (R_d_) according to [34] using non-linear least-squares curve fitting (nlinfit, MATLAB, ver. 2020b, The MathWorks Inc., Portola, CA, USA).

#### 2.3.4. Stomatal Conductance Traits

Similar to the light response curve measurements and using the same set-up of 48 plants (i.e., n = 2, N = 48), stomatal conductance (g_s_) was measured on the abaxial right side of leaf number 13 and 11, for the two lettuce cultivars, using a leaf porometer (AP4, Delta-T Devices Ltd., UK) [35]. The instrument was adapted to the measuring ambient for one hour prior to calibration (±5%), which was performed in the same environment (growth chamber).

#### 2.3.5. Stomata Morphology

Stomatal imprints (n = 2, N = 48) were taken from the abaxial left side of leaf 13 and 11 of the same plants used for leaf conductance readings. Imprints were taken within the growth area and during light with the respective treatments. A fluid silicone (Elite HD+ Super Light Body, Zhermack Dental, Marl am Dümmer, Germany) was spread on the leaf using a dispenser (D2, Zhermack Dental, Germany) to obtain a negative imprint of the leaf lower surface. The fluid was applied instantaneously with minimized physical contact to the plant to avoid measuring related stomata reactions. After hardening of the silicon, a thin layer of transparent nail polish was applied on the silicone imprint to obtain a positive one [36]. 

The latter was photographed in three sections of 133.9 mm^2^ each (total leaf area measured per plant sample = 401.7 mm^2^) at a zoom of 700X (lighting: full coaxal (30%), transmitted (20%)) using a digital 4K microscope (Keyence VHX-7000, KEYENCE DEUTSCHLAND GmbH, Germany). Measurements determined on the images (Figure 2) included stomatal index (stomatal index (%) = (number of stomata/number of stomata + number of epidermal cells) x 100), stomatal density (= number of stomata on the leaf area), stomata length, stomata width, pore length, and pore width [37]. Pore width was adopted to describe the stomatal pore aperture.

### 2.4. Destructive Measurements

At the end of each of the three experimental replications in time (i.e., at 31 DAS and 16 days under experimental light conditions), destructive measurements were performed. Different plants and plant parts were used for various destructive observations as described in the sub-sections below. The work was done between 9:00 a.m. and 5:00 p.m., using a structured sampling protocol (i.e., leaf 12 and 14 were measured for spectral measurements and, immediately after, sampled in liquid nitrogen, rosette excision was conducted at the shoot base for leaf number count, leaf area measurement, and fresh and dry weight determination).

#### 2.4.1. Quantification of Leaf Pigment Content and Estimation of Anthocyanin Content

Two plants per experimental replicate (n = 2, N = 48) were sampled for leaf pigment quantification and estimation. Leaf number 14 (CVg) and 12 (CVr) were used. 

#### 2.4.2. Optical Leaf Measurements and Estimation of Anthocyanin Content

Reflectance was measured on each leaf (both sides of the midrib) using a double-beam spectrophotometer (V-670, Jasco, Japan). Relevant reflectance values were used to calculate mARI and PRIn to estimate leaf anthocyanin content and plant photosynthetic performance, respectively. Indexes were calculated as: mARI = [(R_530-570_^−1^ − R_690-710_^−1^) ∗ R_NIR_] and PRI_n_ = PRI/[RDVI ∗ (R_700_/R_670_)] [38,39,40,41].

#### 2.4.3. Extraction and Quantification of Leaf Pigment Content

Leaf disk samples of each replication were kept at −80 °C and lyophilized and milled in following batches, ensuring immediate extraction after sample processing. The resulting powder of each biological sample was weighed in three technical replicates. After 48 h of extraction in three consecutive washes with 95% ethanol, the obtained extracts were read (at 470, 649, and 664 nm) in triplicates against the same amount of blank solution using a UV/VIS spectrophotometer (Infinite M200PRO, Tecan, Switzerland). The plate was read in a 96-well half area microplate, which was used to ensure a 1 cm pathlength [42].

#### 2.4.4. Growth and Morphology Measurements

Intact plants (n = 4, N = 96) were destructively harvested at 31 DAS and rosette and root fresh and dry weights were determined as described by [42]. For a sub-sample of plants (n = 3, N = 72), the total number of leaves per plant was counted and the area of each leaf was read and summed up (using a leaf area meter, the LI-3100C Area Meter, LI-COR Biosciences, Lincoln, NE, USA) to obtain the total leaf area of each plant.

#### 2.4.5. Data Processing and Statistics

Data were processed and statistically analyzed using Microsoft Excel 2016 and R studio (R version 3.5.2 (20 December 2018), “Eggshell Igloo”) with package “doebioresearch” [43]. Outlier values (range: 0.025–0.975) of each dependent variable were removed prior to statistical analysis. An analysis of variance (ANOVA) test at *p* ≤ 0.05 was applied to the normally-distributed data with a split plot design considering light treatment as the main plot factor, cultivar as the subplot factor, and replication as the block. As a post hoc test, Tukey’s honest significant difference (HSD) test was performed to locate the statistically pairwise comparison between the treatments and cultivars. All measured endpoints were individually analyzed (rosette fresh and dry weights, number of leaves per plant, plant leaf area, minimum (F_0_′) and maximum (F_m_’) chlorophyll fluorescence intensity in the light, maximum operating efficiency of PSII photochemistry in the light (F_v_/F_m_’), stomatal conductance (g_s_), stomata width and length, pore width or aperture and length, stomatal density and index, chlorophyll a and b and their ratio, carotenoids, maximal gross (P_g,max_) photosynthetic rate, normalized photochemical reflectance index (PRI_n_) and modified anthocyanin reflectance index (mARI)).

## 3. Results

Split plot design ANOVA reported that most of the significant differences in the measured variables were between the two lettuce cultivars, Aquino cv. (CVg) and Barlach cv. (CVr), and, to a lesser extent, between the two light treatments, WB and WW (Table 2, Figure 3, Figure 4 and Figure 5).

Major differences between the two cultivars were found in rosette weight, total leaf area, chlorophylls, carotenoids, and PRI_n_. After 15 days of exposure to blue-enriched light, CVr was characterized by a 25% (under WW)–19% (under WB) greater rosette fresh weight compared to CVg, reflecting the faster plant development shown by the red cultivar since seedlings establishment.

Chlorophyll a content was greater (approx. 20%) in CVr and, consequently, chlorophyll a:b ratio was greater (15%) in CVg. For carotenoid content, which was greater in CVr, the difference between the two cultivars was almost doubled under WB light treatment (15% greater carotenoid content in CVr than CVg) compared to WW light control treatment (9%).

A statistically significant effect of the light treatment was found for stomatal conductance and P_g,max_ in both cultivars. P_g,max_ was significantly increased under WB light compared to control light treatment (WW) in both cultivars, though the treatment effect was more pronounced in CVg. Correspondingly, the net photosynthesis response was stronger in CVg compared to CVr; e.g., net photosynthetic rate P_n_ at 1000 µmol m^−2^ s^−1^ PPFD increased from WW to WB by 23% in CVg compared to 18% in CVr (Figure 3). Likewise, greater values of stomatal conductance were measured under WB and in CVg.

An interactive effect between the light treatment and cultivar was detected for stomatal density and mARI. A similar response extent to WB was observed for stomatal density in CVg (36%) and for mARI in CVr (40%) (Table 2).

## 4. Discussion

Light in plant production, especially in closed-type systems, represents a very powerful tool for driving productivity and produce quality towards desired targets and increasing produce commercial value [44]. The light quality requirements of lettuce, the latter being the model plant in IVFs, have been broadly investigated, and discrepancies between distinct cultivars have often emerged [25,45]. As has been seen for other abiotic stresses, divergent responses have been observed between cyanic and acyanic lettuce cultivars: for example, in [46], where the green cultivar was more sensitive to salinity eustress application, or in [47], where the green cultivar was more plastic in regards to its phenolic compound pool in response to nitrogen deficiency. In our case, except for traits that were characteristics of the lettuce cultivar, e.g., fresh weight, pigments, and mARI, we observed analogous adaptation outcomes of the two cultivars to light quality after 15 days of exposure. Moreover, F_v_/F_m_’, which describes the maximum efficiency of energy harvesting open/oxidized PSII reaction centers in light and reflects the imbalance between PSII and PSI stoichiometry, remained unaffected after 15 days of WB application. In studies where F_v_/F_m_’ was monitored over time, it showed a stabilization with time [48]. In lettuce, the red or blue light effect on F_v_/F_m_’ vanished at 32 days of treatment [49]. The comparable F_v_/F_m_’ values measured after 15 days of treatment, together with the lack of light treatment effect on lettuce weights (both fresh and dry), suggested the plants may have adapted to blue light by implementing cultivar-specific strategies [46].

Nonetheless, variety-specific strategies manifested in the two studied lettuce cultivars, helping the plants to adapt to a blue-enriched light environment. If the cultivar-specific strategy adopted by CVr, expectedly, was the increased leaf anthocyanin content, estimated through mARI [42,50], CVg responded to blue-enriched light by increased stomatal density (Table 2, Figure 4). Anthocyanins are known to help reduce the leaf energy load by absorbing excessive photons, especially of blue-green wavelengths. The decreased light absorption in specific wavebands affected by anthocyanins causes adjustments at the light harvesting system level to better match light harvesting to the available light [20]. Changes in stomatal density occur during leaf development, triggered by the light sensing of mature leaves [51], and can be regarded as a slow mid-term process. Stomatal density and stomatal index (but not stomatal aperture) are reported to increase in plants exposed to long-term blue light [52]. In our case, however, stomatal index (i.e., the ratio of the number of stomata to the total number of stomata and epidermal cells) was comparable between the light treatments (WB and WB). This was probably due to a precisely proportional increase in both the number of stomata (% > 51.1) and number of cells (% > 51.5) under WB compared to white light.

Blue light is also reported to stimulate stomatal conductance, g_s_ [53], with potential benefits for evaporative cooling and nutrient translocation [54] and photosynthesis [55,56]. In our case, g_s_ was increased under WB in both cultivars, though a significant effect was only detected in CVg, probably due to the increased stomatal density. The two processes, i.e., increased stomatal density and conductance, tentatively helped to reduce the leaf heat load under WB. Similarly, WB caused an increased capacity of photosynthesis, denoted by an increased theoretical P_g,max_ and higher photosynthesis levels (Figure 3 and Figure 4). Photosynthesis was increased to a greater extent in CVg compared to CVr, reflecting the lower photosynthetic capacity of cyanic leaves [57].

PRI_n_, in the literature proposed as an alternative measure of radiation use efficiency that is also valid across species [38,58], in our case did not correlate with calculated LUE (correlation coefficient: 0.308, *p*-value = 0.329). Calculated PRI_n_ was comparable between light treatments and, in contrast to LUE, it was different in the two studied lettuce cultivars and doubled in CVr compared to CVg. The potential reason behind such cultivar distinction may be found in the greater pigment pool characteristic of cyanic leaves and suggests that PRI_n_ can better describe the foliar photochemistry than the radiation use efficiency.

As disclosed in our initial hypotheses, most of the measurable plant responses after a period of long-term increased radiation energy, in this case applied as 15 days of blue-enriched light application, remained unaffected. Our data suggest that adaptation to a high energy radiation occurred and it was similar in both cyanic and acyanic lettuce cultivars (resulting in increased photosynthetic capacity and stomatal conductance), though with alternative adaptive strategies, i.e., increased stomatal density in the green lettuce cultivar and increased leaf red pigmentation in the red lettuce cultivar, leading to similar growth performance.

## 5. Conclusions

An increased proportion of blue light was used to induce nutritional enhancement (carotenoids and anthocyanins) in two lettuce cultivars; no detrimental effect on growth was observed either in the cyanic or in the acyanic variety. Qualitative and quantitative screening of the secondary metabolites, which was not possible in our case, could represent an interesting investigation for future works. Unexpectedly, long-term effects of blue light did not impact biomass accumulation, though our results reveal the inefficiency of the treatment due to the generated energy waste as, in spite of the greater amount of energy required by WB, no further improvements in rosette weight occurred. Therefore, blue light could be used for short-term application, as recommended in the literature, to enhance nutritional and morphological traits. Moreover, its potential as a hardening treatment, through adjustments in nutritional composition and plant morphology, could be further investigated for improving produce shelf life.

Another interesting aspect is represented by the blue light beneficial effects on stomatal traits and photosynthetic capacity; these blue light specific effects could be further investigated and exploited to trigger increased plant productivity by application of defined light treatments. Moreover, characterization and understanding of cultivar-specific responses to abiotic stresses, e.g., increased stomatal density in response to blue-enriched light in ‘Aquino’ lettuce cv., could generate valuable knowledge in breeding plants for specific environments or purposes.

## Figures and Tables

**Figure 1 biology-11-00959-f001:**
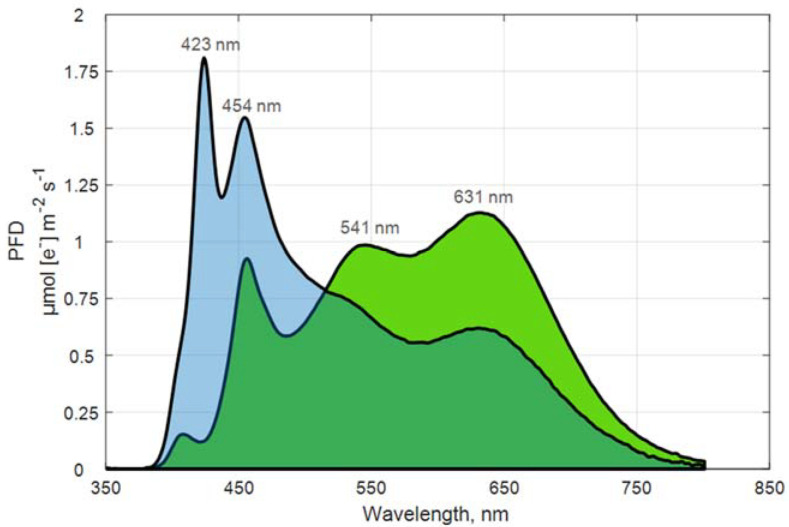
Averaged measured light spectra (average of 3 measurements) for the light treatments tested, blue-enriched white light (WB, in light blue) and white light (WW, in green), with indication of emission peaks.

**Figure 2 biology-11-00959-f002:**
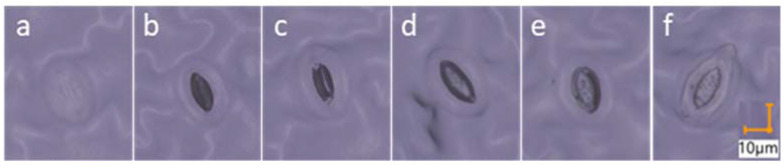
Model images of lettuce stomata positive imprints measured by image analysis using a digital 4K microscope. Images (**a**–**f**) show the increasing opening of the stomatal pore. Images scale: 10 µm.

**Figure 3 biology-11-00959-f003:**
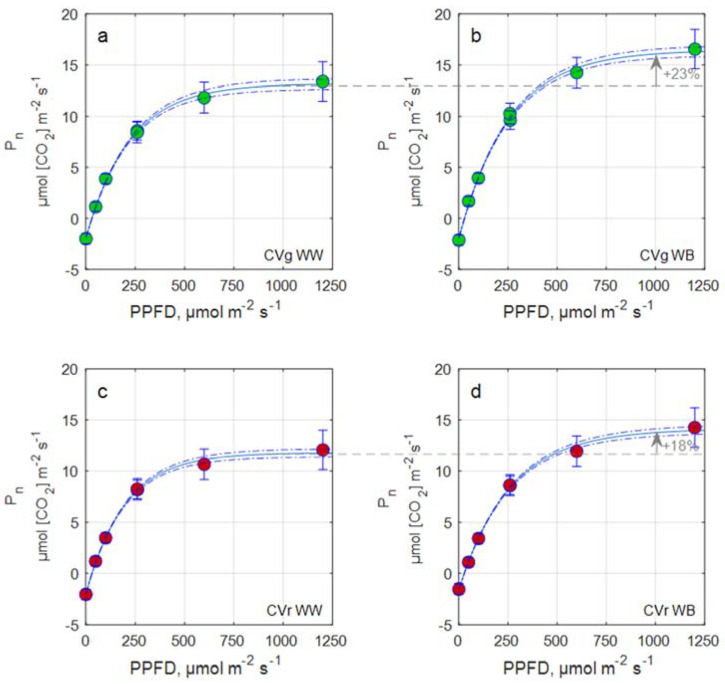
Exponential light response curve with maximum net photosynthesis and chemical light use efficiency (LUE) fitted to measured net photosynthesis rate, P_n_ for Aquino (CVg) (**a**,**b**) and Barlach (CVr) (**c**,**d**) treated with white light (WW) or white-blue light (WB) spectra for 15 days.

**Figure 4 biology-11-00959-f004:**
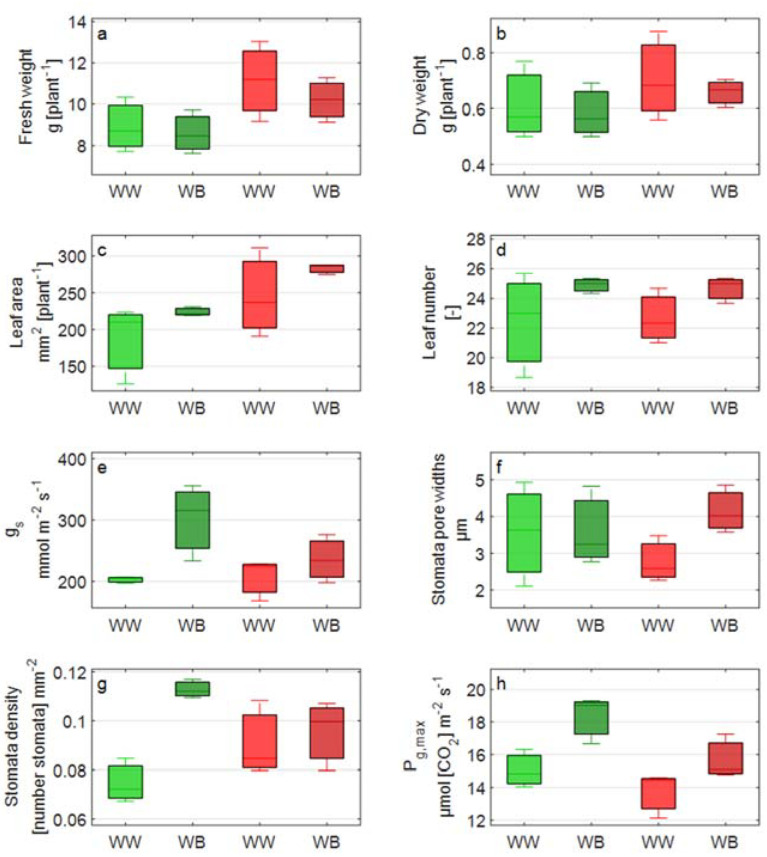
Boxplot overview of the measured variables: (**a**) plant fresh weight; (**b**) plant dry weight; (**c**) plant leaf area; (**d**) leaf number; (**e**) stomatal conductance, g_s_; (**f**) stomatal pore aperture; (**g**) stomatal density; and (**h**) maximal gross photosynthetic rate, P_g,max_. Measurements were taken on the two lettuce cultivars, Aquino (CVg) and Barlach (CVr), treated with white light (WW) or white-blue light (WB) spectra for 15 days.

**Figure 5 biology-11-00959-f005:**
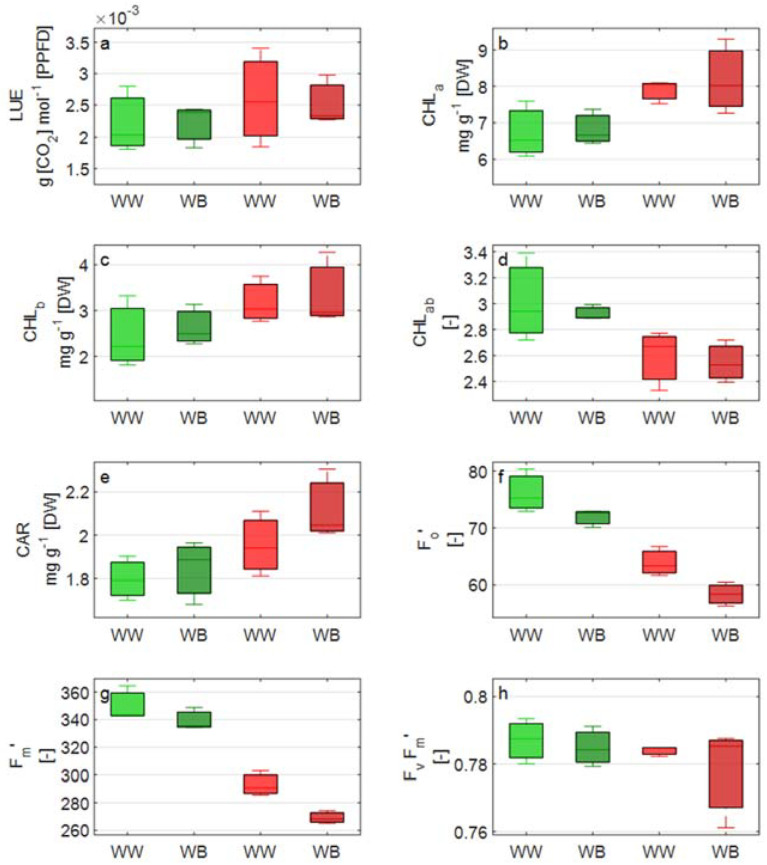
Boxplot overview of measured variable: (**a**) light use efficiency, LUE; (**b**) chlorophyll a content; (**c**) chlorophyll b content; (**d**) chlorophyll a:b ratio; (**e**) carotenoid content; (**f**) minimum value for chlorophyll fluorescence at light, F_0_′; (**g**) maximum chlorophyll fluorescence at light, F_m_’; and (**h**) maximum operating efficiency of PSII photochemistry in the light, F_v_/F_m_’. Measurements were taken on the two lettuce cultivars, Aquino (CVg) and Barlach (CVr), treated with white light (WW) or white-blue light (WB) spectra for 15 days.

**Table 1 biology-11-00959-t001:** Spectral composition (in percentage) of the two light treatments, white light control (WW) and white-blue light (WB), clustered in four main wavelength groups: blue 400–480 nm, green-yellow 481–599 nm, red 600–699 nm, and far-red 670–800 nm, and the indicated light peak (λ_Peak_).

	WW	WB
Blue (400–480 nm)	15	40
Green-yellow (481–599 nm)	40	34
Red (600–669 nm)	29	16
Far-red (670–800 nm)	16	10
λPeak, nm	631	423

**Table 2 biology-11-00959-t002:** ANOVA results based on split plot analysis with light treatment as the whole plot factor, cultivar as subplot factor, and replications as block. Effects of lettuce cultivar (cv. ’Aquino’, RZ, CVg and cv. ‘Barlach’, RZ, CVr) exposed to 15 days of light spectral treatment (blue-enriched white light, WB, and white light, WW) and their interactions on the measured dependent variables: biomass, morphological traits, light-adapted chlorophyll a fluorescence (F_0_′, F_m_’, F_v_/F_m_’), stomatal conductance (g_s_), stomatal pore aperture, stomatal density, stomatal index, chlorophylls, carotenoids, maximal gross (P_g,max_) photosynthetic rate, photochemical reflectance index (PRI_n_), and modified anthocyanin reflectance index (mARI).

Dependent Variables	Replication		Light Treatment		Cultivar		Interaction	
	Df	MS	Df	MS	Df	MS	Df	MS
Rosette fresh weight	2	22.83.	1	11.39 ns	1	69.31 ***	1	3.39 ns
Rosette dry weight	2	0.21 ns	1	0.07 ns	1	0.10 **	1	0.01 ns
Number of leaves	2	19.00 ns	1	44.44 ns	1	0.00 ns	1	0.44 ns
Plant leaf area	2	8969.00 ns	1	12,428.00 ns	1	31,840.00 ***	1	0.00 ns
F_0_′	2	45.92 ns	1	61.91 ns	1	2.70 **	1	2.70 ns
F_m_’	2	443.00 ns	1	134.00 ns	1	38,841.00 ***	1	160.00 ns
F_v_/F_m_’	2	9.55 × 10^−5^ ns	1	4.11 × 10^−5^ ns	1	3.00 × 10^−5^ ns	1	5.20 × 10^−5^ ns
g_s_	2	23,337.30 *	1	16,684.00 *	1	2686.70 ns	1	2.20 ns
Pore aperture	2	5.32 ns	1	0.14 ns	1	2.78 ns	1	10.70 ns
Stomata density	1	0.00 ns	1	0.00·	1	0.00·	1	0.00 **
Stomata index	2	1.64 ns	1	3.32 ns	1	2.24 ns	1	2.23 ns
Chlorophyll a	2	11.61 ns	1	0.96 ns	1	12.50 ***	1	0.02 ns
Chlorophyll b	2	11.01 *	1	0.00 ns	1	4.32 **	1	0.04 ns
Chlorophyll a:b	2	1.60·	1	0.10 ns	1	0.49 **	1	0.01 ns
Carotenoids	2	0.59 ns	1	0.03 ns	1	0.34 **	1	0.05 ns
P_g,max_	2	0.73 ns	1	20.71	1	11.77·	1	1.24 ns
PRI_n_	2	0.00 ns	1	0.00 ns	1	0.00 ***	1	0.00 ns
mARI	2	0.11 ns	1	5.49·	1	85.62 ***	1	5.29 *

Numbers represent degrees of freedom (df) and mean squares (MS). Asterix or ns indicate significant differences at *p* < 0.05, as determined by split plot analysis. Significance codes: 0.000 “***”, 0.00 “**”, 0.01 “*”, ≤0.05 “·”, >0.05 “ns”.

## Data Availability

Not applicable.

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
