# Peer review of "Response of Cyanic and Acyanic Lettuce Cultivars to an Increased Proportion of Blue Light"

_biology, 2022, doi:10.3390/biology11070959_

Round 1

Reviewer 1 Report

The manuscript “Response of cyanic and acyanic lettuce cultivars to increased proportion of blue light” fits the journal’s scope. The authors present their results regarding the influence on two lettuce cultivars of two light treatments with different spectral composition. The design of the study is correct, and the methods and the sampling protocols are presented in sufficient detail. Although the interest to readers may not be high, the methodology used is appropriate, thus the results could be useful for other researchers in future works.

Before publication, some minor corrections/justifications should be made/added.

Conclusions -  are sustained by the authors’ findings. However, the sentence from lines 397-400 should be rephrased.

Introduction - The paragraph regarding the selection of lettuce cultivars should be extended. Thus, the differences and similarities should be better explained.

Please add the missing references for all methods (lines 173, 194, 201, 222, 238, 246).

Author Response

Reviewer #1

The manuscript “Response of cyanic and acyanic lettuce cultivars to increased proportion of blue light” fits the journal’s scope. The authors present their results regarding the influence on two lettuce cultivars of two light treatments with different spectral composition. The design of the study is correct, and the methods and the sampling protocols are presented in sufficient detail. Although the interest to readers may not be high, the methodology used is appropriate, thus the results could be useful for other researchers in future works.

Authors:  We thank Reviewer #1 for his review of our paper and the positive view. We believe and wish that our research could be very interesting for scientists/researchers/engineers in the field of indoor-farming.

Before publication, some minor corrections/justifications should be made/added.

Conclusions -  are sustained by the authors’ findings. However, the sentence from lines 397-400 should be rephrased.

Authors: Thanks for the comment. We have re-phrased the sentence (please see updated document).

Introduction - The paragraph regarding the selection of lettuce cultivars should be extended. Thus, the differences and similarities should be better explained.

Authors:  Thanks for the feedback. We have deepened the information on cultivars differences and similarities in response to the environment (please see updated document).

Please add the missing references for all methods (lines 173, 194, 201, 222, 238, 246)

Authors: Thanks, for noting the absence of references in M&M. We have added all missing references (please see updated document).

Reviewer 2 Report

In the abstract authors should define the area considered for the study (the name should be added). Please reduce some keywords.

In the introduction, the authors well described the focus on the main research topics, and the most relevant questions were addressed. In any case, more information on the influence of agro-environmental conditions on the concentration of molecules should be added. Indeed, literature reported that quali-quantitative amounts of compounds often depend on genotype, soil, plant phenology, used agrotechniques, etc.

In Material&Methods more information on raw materials should be added (information on sampling site and time, used agrotechniques, etc). 

In the result section, the authors could compare their results with other previous studies performed for similar purposes (also considering other species or cultivars). The discussion section could be integrated with the results to avoid repetitions.

The conclusion is clear in relation to the study, but it should be linked in a better way to the other parts of the paper. 

Author Response

Reviewer #2

Authors: We thank Reviewer #2 for his review of our paper and his positive answers.

In the abstract authors should define the area considered for the study (the name should be added). Please reduce some keywords.

Authors: Thanks for the attentive comment. We have reduced the keywords, and included the area of the study as indicated in the adjusted abstract (please see updated document).

In the introduction, the authors well described the focus on the main research topics, and the most relevant questions were addressed. In any case, more information on the influence of agro-environmental conditions on the concentration of molecules should be added. Indeed, literature reported that quali-quantitative amounts of compounds often depend on genotype, soil, plant phenology, used agrotechniques, etc.

Authors: Thanks for suggesting an improvement of our manuscript introduction. We added a new paragraph with more information on the environmental effects on molecules, and the effect of different genotypes (please see updated document).

In Material & Methods more information on raw materials should be added (information on sampling site and time, used agrotechniques, etc). 

Authors: Thanks for remarking this. We have added more information on dates, times and design, we think that now the information provided is complete (please see updated document).

In the result section, the authors could compare their results with other previous studies performed for similar purposes (also considering other species or cultivars). The discussion section could be integrated with the results to avoid repetitions.

Authors: For better understanding of results and discussion, we have chosen to keep the classical structure of two separate sections, each one for Results and Discussion. In the Discussion section, we have proceeded the review of the Introduction and set it in relation to our findings. We have included the main literature on this topic and compared our findings with recent and earlier studies. However, as major parts of our investigations are new, we restricted our comparing discussion to related scientific investigations/papers. In that, we have used the most important and most central literature on our research topic “we investigated the response of cyanic and acyanic lettuce cultivars to increased proportion of blue light”. We are convinced that our literature study is complete.

The conclusion is clear in relation to the study, but it should be linked in a better way to the other parts of the paper. 

Authors: We have improved the Conclusion section according to Reviewer #2 suggestion (please see updated document). 
